# Relevant Indicators of Consciousness after Waterbath Stunning in Turkeys and Stunning Efficiency in Commercial Conditions

**DOI:** 10.3390/ani13040668

**Published:** 2023-02-14

**Authors:** Alexandra Contreras-Jodar, Aranzazu Varvaró-Porter, Antonio Velarde, Virginie Michel

**Affiliations:** 1Animal Welfare Program, Institute of Agrifood Research and Technology (IRTA), 17121 Monells, Spain; 2Direction of Strategy and Programmes, French Agency for Food, Environmental and Occupational Health & Safety (ANSES), 94701 Maisons-Alfort, France

**Keywords:** animal-based indicators, inter-observer repeatability, slaughterhouse, state of consciousness, stunning efficiency, turkeys, waterbath stunning

## Abstract

**Simple Summary:**

Multi-bird waterbath stunning (WBS) is the main commercially used stunning method in poultry. However, not all birds are unconscious after WBS and some of them recover consciousness before death due to differences in individual bird’s resistance to electricity. In order to minimize pain, distress and suffering of ineffectively stunned turkeys, operators must be trained to identify indicators of consciousness so they can re-stun the birds using backup methods. However, the literature reflects that there is no standardization in the applied indicators. In this sense, the prevalence of detected conscious birds can be biased according to the indicators chosen. Relevant indicators must be valid, feasible and repeatable. The validity and feasibility of indicators of the state of consciousness in poultry have been reported by the EFSA (2013) but their repeatability is still a gap of knowledge. Therefore, the aim of the present study was to assess the repeatability between observers of the most valid and feasible indicators of consciousness after waterbath stunning in poultry. This study proposes a refined list of indicators that could be used to assess the state of consciousness of turkeys in commercial slaughterhouses and reports on the efficacy of stunning of different key electrical parameters.

**Abstract:**

The prevalence and inter-observer repeatability of the four most valid and feasible indicators of consciousness after waterbath stunning in turkeys were evaluated before bleeding (i.e., tonic convulsion, breathing, spontaneous blinking and vocalization) and during bleeding (i.e., fluttering, breathing, spontaneous swallowing and head shaking). Furthermore, correlations between them were calculated to further understand the association between such indicators. This study compared the assessments of four observers on 7877 turkeys from 28 flocks slaughtered in eight different abattoirs. Repeatability between observers was assessed by combining the crude percentage of agreement and the Fleiss’ kappa. Before bleeding, tonic seizure was observed to be not reliable under commercial conditions and spontaneous blinking and vocalization had poor repeatability. During bleeding, spontaneous swallowing was not observed and head shaking had poor repeatability. Hence, the most relevant indicators of consciousness before bleeding is breathing while during bleeding is breathing and wing flapping. Moreover, the stunning efficiency of the key electrical parameters used in the different slaughterhouses was assessed. Therefore, a refined list of indicators of the state of consciousness after waterbath stunning is proposed to be used in commercial turkey slaughterhouses to ensure consistency of controls. On the other hand, the range of the prevalence of turkeys with indicators of consciousness within a batch found before bleeding [0–16%] and after bleeding [0–18%] highlights the importance of controls.

## 1. Introduction

Current European Union legislation on welfare of animals at the time of killing requires stunning before slaughter to render the animals unconscious, and thus insensible to pain, until death occurs [1].

Although several stunning methods are used for turkeys [1], the main stunning method commercially used is by far electrical waterbath stunning (WBS) followed by controlled atmosphere stunning (CAS) [2].

WBS involves inverting and hanging live birds by their legs in metal shackles onto a conveyor line that transports them to a waterbath, where multiple birds are immersed simultaneously up to the base of their wings in electrified water. The WBS system contains two electrodes, one placed at the bottom of the waterbath and the other on the shackle line. The electrical circuit is completed when the birds head and neck contact the electrified water so the current passes through the bird’s head and body. The bird is immediately unconscious and insensitive to pain, if sufficient current passes through the brain [3]. Disadvantages of this method include that birds experience stressful and painful shackling, the potential risk of receiving pre-stun electric shocks and delayed induction of unconsciousness. In addition, runts are more prone to miss the WBS and be conscious at the time of bleeding [4]. Finally, WBS is not always effective despite using the key electrical parameters established by the EU legislation. In this sense, not all birds are unconscious after WBS and some of them recover consciousness before death due to differences in individual bird’s resistance to electricity. Resistance is variable according to several factors such as age, size, sex [5,6,7], feather coverage and condition (wet or dry) [8], fat-to-muscle ratio and hydration status [9]. On the other hand, CAS is a relatively new method that consists of using gas mixtures to render birds unconscious, avoiding many of the welfare problems associated with the electrical WBS. Allowed CAS use carbon dioxide gas only; some use inert gases and others use a mixture of inert gases and carbon dioxide. Those using carbon dioxide expose birds to one or more ‘phases’, whereby each phase contains a different concentration of the gas being used: starting at a low concentration (i.e., <40%) and increasing to a higher concentration when birds are unconscious to ensure death [10]. Depending on the CAS system design, the gas selected, and the concentration of the gas, this process can cause aversive behaviors before the loss of consciousness (respiratory distress, escape attempts) or render the birds unconscious rapidly and humanely. CAS is gaining ground year by year although WBS is still in use in both small- and large-scale turkey slaughterhouses (SHs). This is mainly because the investment required for CAS is much higher but also because CAS is generally not accepted for religious slaughter while WBS is for some Muslim communities, provided that birds are reversibly stunned by this method [11].

Since WBS is not always effective, it is mandatory in the European Union that business operators regularly monitor the state of consciousness of birds after stunning. Hence, birds that show indicators of consciousness should be re-stunned using backup methods to avoid them unnecessary pain, distress and suffering [1].

In 2013, the EFSA [12] pointed out that birds stunned by means of electrical waterbath should be monitored before and during bleeding using animal-based indicators (ABIs). In addition, they reported a list of ABIs ranked according to the validity (i.e., sensitivity) and feasibility perceived by a panel of experts. Those had the highest validity and feasibility were shown as recommended indicators while the others were kept as optional. It should be highlighted that it was addressed for poultry but not species specific. However, information on which indicators of consciousness are the most commonly observed in each poultry species and the repeatability between observers are not yet explored.

A study conducted in Belgian poultry abattoirs reflected that monitoring the state of consciousness after WBS was not always performed; and when it was, there was no uniformity in the applied criteria [13]. Moreover, several scientific studies assessed the efficiency of stunning according to different key electrical parameters in the waterbath. However, inconsistency of method and criteria was also observed [14,15,16].

Thus, it is of utmost importance to gain insight into which are the most relevant ABIs for monitoring the state of consciousness in poultry and select them, ensuring consistency of controls. Recently, a refined list of ABIs for broiler chickens was proposed [17] but no refined list exists for any other poultry species.

The main goal of the present study is to assess the repeatability between observers and the prevalence of the four most valid and feasible ABIs before and after bleeding according to the EFSA [12] as well as the correlation among them. Taking everything into consideration, we offer a refined list of the most relevant and reliable ABIs so that they can be used to evaluate the state of consciousness of turkeys in commercial SHs ensuring consistency of controls. On the other hand, additional goals include the evaluation of the effectiveness of stunning according to different combinations of key electrical parameters of the waterbath (frequency and current) used in different commercial SHs and batches of turkeys.

## 2. Materials and Methods

### 2.1. Selection of Slaughterhouses and Animals

Eight commercial turkey SHs equipped with WBS were selected across France and Spain. Selection of the SHs was carried out in conjunction with the official veterinary services and reflects a certain diversity in terms of plant design, electrical key parameters, turkey genotypes, and line speed. Each slaughterhouse was assigned to a number to keep their anonymity (SHs from 1 to 8).

### 2.2. Description of the Slaughterhouses and Waterbath Stunning Systems

Substantial variation in SH design, WB length and capacity, line speed, stun-to-stick interval and bleeding method was observed between the SHs assessed. The main characteristics are shown in Table 1.

Line speed was not measured in situ but was reported by the food business operators and the official veterinary service. Line speed was higher in the slaughtering of female turkeys and so the exposure time in the waterbath was lower compared to the male turkeys in SH-2. This was because females slaughtered in SH-2 weighed approximately 6–8 kg and males approximately 16 kg; and therefore, line speed in females can be increased since the time needed to eviscerate, wash and dress the carcasses is lower than in males.

A digital control panel monitored the electrical parameters applied (i.e., total current passing through the WB, voltage and frequency) in all SH. The automatically recorded electrical parameters were obtained from the SH but were not measured and verified. The average values of current per animal were calculated by dividing the total current amount passing through the WB by the number of birds simultaneously submerged in the water and they are shown in Table 2. The electrical waveform was sine alternating current in all SHs. The bleeding was made by severing the carotids, but the procedure differed among SHs: seven performed bleeding by neck incision (SH-1, SH-2, SH-3, SH-4, SH-7, and SH-8); while the other two performed it by oropharynx incision (SH-5 and SH-6). All SHs performed manual bleeding except for SH-7 that set up blades that produces automatically bilateral neck cut. Slaughter line speeds ranged from 300 to 3000 turkeys/h.

### 2.3. Assessment of the Consciousness

#### 2.3.1. Observers

The assessment of turkeys’ state of consciousness was carried out by four trained observers that conducted a similar study in broiler chickens (using the same ABIs for the state of consciousness) with an overall κ value of 0.77 [17]. Each observer was named as letter (A to D). An additional person randomly selected the birds to be assessed by pointing at the bird’s breast with a laser pointer. This was aimed at helping to track the animal and prevent mistakes at evaluating all four observers the same selected bird. The assessment of consciousness was conducted in a representative sample of birds in two different places of the slaughter line: (1) from the exit of the WB until bleeding and (2) at approximately 10 s after severing the carotids (Figure 1). The four observers evaluated each bird for 6 to 8 s (depending on SH design and visibility) and scored the ABIs without discussing or disclosing their assessments during the evaluation. Line speed was not reduced in any case to facilitate the observers’ assessment.

#### 2.3.2. Sample Assessment

Different batches of turkeys were assessed. The evaluation of the state of consciousness of each batch was carried out by alternating the observation of 50–100 turkeys before bleeding and after bleeding until the whole batch had been slaughtered.

Sometimes an observer was distracted for whatever reason (e.g., business operators passing in front of them) and could not assess the turkey that was being indicated with the laser. In these cases, the observers made a note in their observations and the outcomes of the other observers were filtered out for repeatability assessments.

A summary of the electrical parameters used for each batch and SH, along with the characteristics of the turkeys in the batch and the number of assessed birds, is shown in Table 2. Each batch was composed of either males or females but with exceptions. SH-4 and SH-5 slaughtered batches of females, but some males were found mixed in the slaughter line. Similarly, SH-8 slaughtered a batch of males although some females were encountered mixed in the slaughter line.

#### 2.3.3. Indicators for the Assessment

The most valid and feasible ABIs for the assessment of the state of consciousness before and during bleeding were selected from those identified by the EFSA [12]. The selected ABIs before bleeding were tonic seizure, breathing, spontaneous blinking and vocalizations, while the selected ones during bleeding were wing flapping, breathing, spontaneous swallowing and head shaking. The description of the outcomes of consciousness and unconsciousness of these ABIs are those reported by Contreras-Jodar et al. [17] and are summarized in Table 3.

The four observers standardized the protocol by agreeing beforehand on the description of the indicators, the assessment methodology, and the score. Then, they placed where there was the best possible visibility towards the shackled birds from a ventral position. Nevertheless, the turkeys were occasionally evaluated dorsally at the exit of the WB and during bleeding rather than ventrally (in SH-2, SH-3 and SH-5) or before but not during bleeding (SH-6) due to variability in the design and layout of the SHs. When it occurred, the assessment of breathing by direct observation of abdominal muscles around the cloaca was impaired. Data were recorded as binomial as 0 if the outcome of unconsciousness was observed and 1 when an outcome of consciousness was observed. The presence of at least one outcome of consciousness indicates the risk of the bird being conscious or regaining consciousness after WBS and therefore of an ineffective stunning or a long stun-to-stick interval (i.e., time from stunning to the start of bleeding).

### 2.4. Statistical Analysis

Data pre-processing, statistical analyses and plots were performed using R software v.4.1.0. [18]. First, birds that were not assessed by all four observers were filtered out to ensure that all observations were directly comparable. For all the statistical analyses, significance was declared at *p* < 0.05.

#### 2.4.1. Inter-Observer Repeatability of Animal-Based Indicators

The overall level of agreement between observers for each ABI were determined and expressed by two values: the crude proportion of agreement (PoA) and the Fleiss’ kappa (κ). Both were computed using the “irr” package of R software [19]. The PoA can be misleading, as it does not take into account the scores that the observers assign due to chance. This flaw is solved by computing the κ since it expresses the extent to which the observed concordance between observers exceeds the proportions that would be expected if all observers scored completely at random. κ ranges from −1 to +1, with −1 representing perfect disagreement between observers, 0 indicating the amount of agreement one can expect given a random chance and 1 representing perfect agreement between observers [20]. In addition, κ is a standardized value so it is interpreted in the same way across multiple studies. Hence, κ can be classified as “excellent” agreement beyond chance if values are greater than 0.75; “fair to good” agreement beyond chance if values between 0.40 and 0.75 and “poor” agreement beyond chance if the values are below 0.40. Nonetheless, when there is an insufficient scoring variation in the evaluated indicator (i.e., low prevalence of indicators of consciousness), although high inter-observer agreement, κ appears close to 0 [21].

#### 2.4.2. Correlation among Animal-Based Indicators

The chi-squared % defective test was used to determine if there were statistical differences (divergence) among observers between the expected and the observed frequencies of every outcome of consciousness of the evaluated indicators. If one observer differed statistically from the others at evaluating the ABIs, the mean of the proportion of the closest evaluations or the in between value when scoring were not consistent among them were recorded. Since data did not follow normal distribution, associations between observed ABIs of consciousness was performed using Spearman’s rank correlation test. Correlation results were presented as heat map. Proportions among combinations of ABIs were performed as a Venn diagram considering all turkeys assessed in the present study using the “eulerr” package [22].

#### 2.4.3. Relationship between Electrical Parameters and Stunning Efficiency

Stunning inefficiency of each batch and SH was evaluated by showing the percentage of birds with at least one outcome of consciousness in any of the stages of the assessment: before and during bleeding. The chi-squared % defective test was used to determine if there were statistical differences among observers between the expected and the observed frequencies of every outcome of the indicators evaluated. If one observer differed statistically from the others at evaluating the ABIs, the mean of the proportion of the two closest evaluations or the in between value when scoring were not consistent among them were reported. The prevalence of each indicator within each batch was calculated and from this, the interval of confidence in the population (here in each batch) was calculated. Thus, 95% confidence interval of turkeys showing outcomes of consciousness was computed using the Wilson’s formula from “epitools” package of R software [23] in every batch assessed.

## 3. Results

### 3.1. Inter-Observer Repeatability of the Animal-Based Indicators

#### 3.1.1. Before Bleeding

Four ABIs of the state of consciousness were assessed before bleeding. These were tonic convulsion, respiration, spontaneous blinking and vocalization. The mean prevalence of outcomes of consciousness per batch of birds, per observer and SH is shown in Table 4. Furthermore, the global level of agreement between the four observers for the aforementioned ABIs per SH is shown in Table 5.

##### Tonic Seizure

Absence of tonic seizure was observed in all SHs although with different prevalence depending on the SH assessed (Table 4). While SH-3 did not exceed the 17.2% in average among observers, SH-1 and SH-4 had the highest prevalence of absence of tonic seizure in the sample (100%). In any case, the PoA was higher than 76% in all but one of the eight slaughterhouses. SH-2 had the lowest PoA (58.6%) as observer C scored 1.6 times more birds with absence of tonic seizure when compared to the other observers (*p* < 0.001). The prevalence of absence of tonic seizure differed considerably between SHs and, therefore, the κ and its interpretation where SH-2 was interpreted as “poor agreement”; SH-3, SH-5 and SH-6 as “fair to good” and SH-7 and SH-8 as “excellent”. κ could not be computed neither in SH-1 nor SH-4 due to absence of scoring variation as all turkeys were assessed with absence of tonic seizure (Table 5).

81.2% of the total number of turkeys assessed (n = 3690) showed absence of tonic seizure (Table 4) and the PoA between observers was 88.4% and the κ was statistically significant and interpreted as “excellent” (κ = 0.79; *p* < 0.001; Table 5).

##### Breathing

Turkeys with presence of breathing was observed in all SHs assessed. The highest prevalence of breathing in a sample was found in SH-5 with an average of 8.5% (Table 4). The PoA was above 98.4% in all SHs (Table 5) and there was no divergence on scoring among observers in any SH (*p* > 0.05) (Table 5). However, there was divergence of κ linked to the different degree of the prevalence of breathing among SHs (Table 4).

Taking all birds from the SHs assessed into consideration, divergence at scoring was found since observer C scored 1.4 times more birds with presence of breathing compared with the other observers (*p* < 0.05). Thus, the calculated average with the closest outcomes for presence of breathing was 1.4% of birds (Table 4). The PoA among observers was high (98.6%) and the κ was statistically significant and interpreted as “excellent” (κ = 0.75; *p* < 0.001; Table 5).

##### Spontaneous Blinking

Turkeys showing spontaneous blinking were observed in all SHs except SH-6 (Table 6). However, the higher prevalence in a sample was found at SH-8 with an average of 0.5% of the turkeys (Table 4). The PoA was above 97.6% in all SH (Table 6) but there was divergence on rating among observers at SH-2 where observer A detected presence of spontaneous blinking in 6 out of 349 turkeys, whereas observer B only observed 1 while observers C and D observed none of them (*p* < 0.01) as shown in Table 4. Moreover, there was divergence of κ and, in most of the SH (SH-1, SH-2, SH-3, SH-4 and SH-5), κ was close to 0. Therefore, the calculated prevalence of spontaneous blinking was low and when present, there was not always consensus among observers. However, κ was interpreted as “fair to good” in SH-7 and SH-8 because when present, the observers agreed in most of the cases (Table 5).

When considering all turkeys assessed before bleeding, lack of consistency at scoring was also found since observer A and B scored 3 times more birds with presence of spontaneous blinking compared with the observers C and D (*p* < 0.05). Therefore, the calculated average was 0.2% (Table 4). The PoA among observers was 99.4% and the κ was statistically significant but interpreted as “poor” agreement (κ = 0.10; *p <* 0.001; Table 5).

##### Vocalization

Vocalization was heard only in SH-5 but if it happened not all the observers agreed (Table 4). Thus, among all ABIs assessed before bleeding, vocalization was the one with the highest PoA (above 99.0%) and there was no divergence on scoring among observers (*p* > 0.05) in any of the SHs assessed.

Taking all birds assessed in this study into consideration, presence of vocalization was extremely low (0.04%; Table 5). Hence, the PoA among observers was 99.9% but the κ was not statistically significant (κ = 0.00; *p* = 0.512; Table 5).

#### 3.1.2. During Bleeding

Four ABIs were evaluated during bleeding: wing flapping, breathing, spontaneous swallowing and head shaking. The prevalence of birds showing outcomes of consciousness, by observer and SH is shown in Table 6 and the overall level of agreement between the four observers according to the SH is shown in Table 7.

##### Wing Flapping

Birds with presence of wing flapping were observed in all SHs except for the SH-1 and the highest prevalence was found in SH-2 (5.1% turkeys in average). In addition, there was uniformity on rating among observers in all the different SHs assessed (*p* > 0.05; Table 6). The PoA among observers was above 94.9% in all SHs assessed (Table 7). κ could not be computed neither in SH-1 nor SH-7 as the PoA was 100%, whereas in the rest of the SHs, κ was interpreted as “excellent” in most of the SHs assessed (Table 7).

Taking all birds from the SHs assessed into consideration, the prevalence was 2.3% and the detection of wing flapping did not differ statistically among evaluators (*p* > 0.05; Table 6). Furthermore, the PoA among observers was high (98.4%) and the κ was statistically significant and interpreted as “excellent” agreement among observers (κ = 0.79; *p* < 0.001; Table 7).

##### Breathing

Birds with presence of breathing during bleeding were observed in all SHs assessed except in SH-1. There was no divergence on scoring presence of breathing among observers in any SH assessed (*p* > 0.05) and highest prevalence occurred in SH-6 (8.8%), followed by SH-4 (6.7%), SH-2 (5.0%), SH-5 (1.9%), SH-7 (1.1%), SH-3 (0.4%) and SH-8 (0.4%) as shown in Table 6. The PoA among observers was above 95% in all SHs assessed (Table 7). κ could not be computed in SH-1 as no bird assessed breathed during bleeding, but this outcome of consciousness offered κ interpreted from “fair to good” to “excellent” (Table 7).

Considering the data from all SH, the mean prevalence was 2.3% and the detection of breathing did not differ statistically among evaluators (*p* > 0.05; Table 6). Moreover, the average PoA was 98.6% and the κ was statistically significant and interpreted as “excellent” agreement among observers (κ = 0.85; *p* < 0.001; Table 7).

##### Spontaneous Swallowing

Spontaneous swallowing was observed only in SH-2 in one animal and only by one of the four observers. Therefore, there was no divergence at scoring spontaneous swallowing among observers (*p* > 0.05; Table 6). For this reason, in SH-2 the PoA was 99.8% but the κ interpretation was classified as “poor” and not statistically significant (κ = 0.00; *p* > 0.05). In the other SH assessed, PoA was 100% and the κ could not be computed due to absence of birds showing this outcome of consciousness (Table 7).

When considering all the birds assessed, the prevalence of spontaneous swallowing was 0.0% (Table 6), the PoA was very high (99.9%), but the κ was not statistically significant and classified as “poor” agreement among observers (κ = 0.00; *p* = 0.504; Table 7).

##### Head Shaking

Birds showing head shaking were observed in all SHs assessed except in SH-1 and SH-8. There was no divergence on scoring presence of head shaking among observers in any SH assessed (*p* > 0.05) and the highest prevalence occurred in SH-2 (1.0%), followed by SH-4 (0.4%), and SH-3, SH-5, SH-6 and SH-7 (0.1% each) as shown in Table 6. The PoA among observers was above 97.8% in all SH assessed, the κ could not be computed neither in SH-1 nor SH-8 as no bird assessed shake its head. In the rest of SH assessed this outcome of consciousness offered κ values interpreted from “poor” in most of cases to “fair to good” (Table 7).

When taking all the birds assessed into consideration, the prevalence of head shaking was 0.2% and the scoring of breathing did not differ statistically among evaluators (*p* > 0.05; Table 6). The PoA among observers was very high (99.5%) and the κ was interpreted as “poor” but statistically significant (κ = 0.29; *p* < 0.001; Table 7).

### 3.2. Correlation among Animal-Based Indicators

#### 3.2.1. Before Bleeding

The proportions of turkeys showing outcomes of consciousness and their combinations of ABIs at the same bird is shown as a Venn diagram (Figure 2a).

Absence of tonic seizure was the most frequent indicator followed by presence of breathing and spontaneous blinking. Vocalization was considered absent in all turkeys assessed since only one out of the four observers detected presence of vocalization in three different turkeys. Combinations of more than one outcome of consciousness were almost non-existent at this stage. Heat map was not displayed in the report as no correlation was found among any ABI.

#### 3.2.2. During Bleeding 

The number of turkeys that showed each outcome of consciousness or each combination of more than one outcome of consciousness is displayed as a Venn diagram in Figure 2b. Thus, wing flapping was the most commonly observed outcome of consciousness, followed by breathing and to a lesser extent, head shaking, whereas no turkeys showed spontaneous swallowing. Additionally, some birds showed breathing accompanied primordially by wing flapping but rarely by head shaking.

Unlike before bleeding, during bleeding there was correlation among the ABIs as shown in Figure 3. All correlations were positive, but only the presence of breathing and wing flapping (r = 0.389) and wing flapping and head shaking (r = 0.391) were statistically significant (*p <* 0.05).

### 3.3. Relationship between Electrical Parameters and Stunning Effectiveness

Stunning effectiveness was analyzed in relation to different combinations of electrical parameters applied to batches of different characteristics (e.g., body weight). To gain some insight into the relationship between combination of electrical parameters and stunning effectiveness and maintenance of the state of unconsciousness in turkeys, the occurrence of turkeys showing indicators of consciousness were compared. Batch 4 from SH-7 was excluded in this section as there was a prevalence of conscious birds in 1.8% of turkeys before and 9.4% of turkeys during bleeding caused by waterbath breakdown but not because of the electrical parameters in use.

#### 3.3.1. Before Bleeding

Most of the turkeys did not exhibit tonic seizure (2941 out of 3659; 80.4%). In addition, this outcome of consciousness was not correlated to others. In these cases, tonic seizure might be previously occurred while the turkey was submerged in the waterbath. It is presumed that it can occur for several reasons, either because of high length waterbath or in low line speeds or because the key electrical parameters were adjusted to stun-to-kill the turkeys. Thus, this indicator has low reliability in commercial conditions, and it will not be considered as an indicator of consciousness when computing the prevalence of conscious turkeys before bleeding. Apart from breathing, the assessment of wing flapping at this stage is highly recommended (although not assessed in the present study) since some turkeys that missed the waterbath appeared wing flapping at the exit. Thus, a summary of the different electrical parameters applied and the 95% confidence interval (95% CI) of birds showing at least one outcome of consciousness before bleeding is summarized in Table 8. The prevalence of turkeys showing indicators in batch 1 of SH-1 is not shown since no turkeys could be assessed before bleeding. This was because the observers began the assessment of the state of consciousness after bleeding, but no turkeys were available for evaluation before bleeding because that flock had already been completely slaughtered.

Ineffective induction to unconsciousness was observed in some animals in 14 out of 22 batches assessed. The combination of electrical parameters that resulted in effective induction to unconsciousness was found in SH-1 (batch 1), SH-2 (batch 4), SH-4 (batch 1) and in all batches assessed in SH-6 and SH-7. Meanwhile, SH-5 showed the highest prevalence in a sample with 5.8% of conscious birds in batch 1 and 15.9% in batch 2.

#### 3.3.2. During Bleeding

The different electrical parameters applied and the 95% CI of birds showing at least one outcome of consciousness during bleeding is summarized in Table 9.

Effective stunning was observed in all batches assessed in SH-1 and in batch 2 and 3 in SH-7 since no turkey showed indicators of consciousness. Meanwhile, batch 1 in SH-7 kept 1% average of turkeys with outcomes of consciousness with a range from 0.3 to 3.6%.

Others that failed at preventing turkeys recovering consciousness were SH-2, SH-3 (in batch 6 and 7), SH-4 and SH-5 and resulted in more than 5% of turkeys with outcomes of consciousness.

## 4. Discussion

The objective of the present study was to gain insight into the prevalence and the repeatability between observers of the most valid and feasible ABIs for assessing the state of consciousness after WBS in turkeys. Additionally, to report on the prevalence of failure to induce and maintain unconsciousness in commercial slaughterhouses.

This study compares the assessment of four observers in 7877 turkeys from 28 batches of eight SHs. SHs were from two turkey producer countries in Europe and despite of not being selected randomly, still represent a wide variety of SH designs, key electrical parameters applied, line speeds and turkey’s body weight. Regarding the observers, all of them were well trained, conducted a similar study in broiler chickens with a κ value of 0.77 [17] and agreed on the definition of the ABIs before the assessments. The number of observers was kept to the maximum number possible with the intention of causing minimum interference to the operators and to each other. For this purpose, they were placed side by side assessing the same turkeys in the same span of time.

### 4.1. Inter-Observer Repeatability of Animal-Based Indicators

Inter-observer repeatability was analyzed per individual assessed using the combination of PoA and κ. Although an outcome of consciousness with a high PoA may suggest that there is high inter-observer repeatability, it may be high because the outcome is very clear for all to detect when present (e.g., presence of wing flapping), because the outcome rarely occurred (e.g., presence of spontaneous blinking and head shaking) or it was hardly ever observed (e.g., presence of spontaneous swallowing and vocalizations). On the contrary, the agreement is lower in the outcomes of consciousness that are more frequently observed (e.g., absence of tonic seizure). On the other hand, the κ interpretation slightly varied according to the SH assessed for most of the indicators. It happened because κ values are strongly influenced by the prevalence of birds showing outcomes of consciousness, and this differed strongly among SHs (when the prevalence is low so is the κ). Spontaneous blinking was the only exception where inter-observer repeatability was interpreted as “poor” in five of the eight SHs, whereas, in vocalizations and spontaneous swallowing, the κ was not able to be computed in seven out of the eight SHs due to absence of this event. These results suggest that these are cases in which the calculation of PoA and κ does not give much information per se while the combination of both do.

Inter-observer repeatability of ABIs for the state of consciousness after WBS in turkeys is “excellent” or “poor” depending on the ABI assessed. The most repeatable ABIs before bleeding are vocalization and spontaneous blinking, followed by breathing and tonic seizure. However, spontaneous blinking and vocalization were artificially highly repeatable because they were hardly ever observed. When occurred though, there was no consensus among the observers. For this reason, vocalization and spontaneous blinking are not considered relevant indicators at this stage while tonic seizure and breathing are, although less repeatable between observers.

During bleeding, the most repeatable indicators are spontaneous swallowing followed by head shaking, breathing and wing flapping. However, spontaneous swallowing and head shaking were artificially highly repeatable because were observed very few times. For this reason, we recommend keeping breathing and wing flapping as relevant ABIs during bleeding despite of being a bit less repeatable. Head shaking could be placed in a second level of attention while spontaneous swallowing could be not even considered. When comparing these results with those obtained previously in broiler chickens [17], the same indictors are recommended before bleeding in both poultry species. However, spontaneous swallowing was observed in very few broiler chickens but never in turkeys while head shaking was considered as a relevant ABI in broiler chickens, but it is not in turkeys during bleeding. In addition, it seems that turkeys are much more prone to breath or flap their wings rather than shake their heads while regaining consciousness. For this reason, this slight difference might raise the necessity to elaborate a list of relevant ABIs that can be specific for each poultry species. Repeatability may be influenced by impaired visibility towards the animal because of the SH design and not all observers could take the most appropriate place for the assessment. Other reasons may be that when paying attention to a specific ABI, the evaluator is more prone to miss either the presence of another outcome of consciousness or the assessment of the other ABIs. However, it is likely that higher levels of inter-observer reliability could be achieved when the observer is alone and can take the best place to perform the assessment and when standardizing descriptions, training and wider testing at assessing consciousness of turkey at slaughter. Thus, better training looks to be one of the key points to improve repeatability.

The prevalence and inter-observer repeatability of the ABIs for the state of consciousness in turkeys slightly differed from those found in broiler chickens [17]. In this sense, the prevalence and repeatability of absence of tonic seizure was higher in turkeys (82.3% and κ = 0.79) than in broilers (7.5% and κ = 0.64). The same occurred in presence of breathing where the prevalence and repeatability in turkeys (1.4% and κ = 0.75) were higher than those reported in broiler chickens (0.9% and κ = 0.58). Spontaneous blinking and vocalization had very low prevalence (≤0.2% and <0.1%, respectively) and “poor” repeatability (κ <0.15) in both turkeys and chickens. After bleeding, the prevalence and repeatability of wing flapping were also higher in turkeys (2.3% and κ = 0.80) than in broilers (1.6% and κ = 0.66). However, breathing was observed to be less frequent in turkeys but with greater repeatability (2.3% and κ = 0.82) than in broiler chickens (13.6% and κ = 0.64). However, head shaking was much less observed in turkeys and had a lower repeatability (0.2% and κ = 0.29) than in broiler chickens (3.4% and κ = 0.64) and spontaneous swallowing never occurred in turkeys while in broilers the prevalence found was 0.7%. The higher repeatability in most indicators may be influenced by the lower line speed range found when assessing turkeys (300 to 3000 turkeys/h) compared to those found in broiler chicken SHs (200 a 10,500 chickens/h). The lower the line speed, the better (and more repeatable) is the assessment of the animal’s state of consciousness.

### 4.2. Correlation among Animal-Based Indicators

It should be highlighted that most turkeys showed absence of tonic seizure (2941 out of 3659 turkeys). In addition, this indicator was not correlated with any other indicator of consciousness before bleeding. It might be caused because tonic seizure occurred while the bird was submerged in the waterbath, due to the longer time spent there when they are long or when the line speed is low. Considering these arguments and the results of our study, tonic seizure is not reliable under commercial conditions. These results are in accordance with those found in broiler chickens under commercial conditions [17].

Given all this, breathing is the most relevant ABI before bleeding.

However, although wing flapping was not included as recommended ABI before bleeding in the EFSA list [12], this indicator should also be considered since some turkeys appeared at the end of the waterbath making efforts to stand upright (avoiding contact with the electrified water) and/or wing flapping because they were not stunned at all (e.g., females found in batches of males).

Sometimes, during bleeding, more than one indicator of consciousness was observed. The most common indicators of consciousness during bleeding were wing flapping and breathing. It seems that when a turkey starts breathing is more prone to show wing flapping soon after but not head shaking. However, when a turkey flaps its wings is more prone to shake its head. Although the correlations were positive and statistically significant, correlation coefficients were rather weak. This could be explained due to the large differences found in the stun-to-stick interval between the different commercial SHs designs. When observing turkeys for a time span of 6 to 8 s at a distance corresponding to 10 s from the neck cutting of the animal, it could be that in short stun-to-stick intervals, breathing or both breathing and wing flapping were the most observed indicators while in longer stun-to-stick intervals, breathing and wing flapping or only wing flapping are the most observed indicators.

### 4.3. Relationship between Electrical Parameters and Stunning Efficiency

One of the objectives of the present study was to compare the effectiveness of stunning between different SHs and different key electrical parameters in waterbath. Results showed that some combinations of the electrical parameters were considered effective at inducing and maintaining the state of unconsciousness during bleeding, and some did not. It should be highlighted that non-effective sub-optimal electrical parameters or equipment, the use of low voltage/current or/and the application of high frequencies and poor or lack of calibration may represent an animal welfare hazard by not guaranteeing the unconsciousness of all animals [16]. Thus, precaution when comparing the efficiency of the electrical combinations is needed as the electrical parameters used in waterbath were obtained from the FVO and where not checked in situ by the researchers in the present study. Additionally, other parameters varied between SHs such as turkey’s body weight, electrode length, distance from head to the electrode, and wetness of shackles that also influence the effectiveness of stunning. Distance from the exit of the waterbath until neck cutting also varied between SHs so delayed bleeding may increase the prevalence of birds that recover consciousness.

It is known that low frequencies (e.g., 75 Hz) are more effective at stunning and require lower current intensities [14]. However, different combinations with higher frequencies can also guarantee bird’s welfare at slaughtering but animal’s body weight and sex should be considered when applying these combinations. Although key electrical parameters were in compliance with current EU legislation, induction of unconsciousness failed in several turkeys in almost all commercial abattoirs evaluated. However, the highest prevalence of conscious animals in a sample was found in SH-5 (batch 1: 5.8%; batch 2: 15.9%). Despite of using similar key electrical parameters in similar average body weights in both batches, the high prevalence observed was due to misplacement of males in batches of females. It looked like the electrical parameters in use were appropriate for inducing unconsciousness in the females (lower body weight) but not for the males (higher body weight). Cases of success were found in SH-1, where all turkeys of approximately 4 kg of body weight were effectively induced to unconsciousness using sine alternative current (AC), 343 ± 41 mA/bird, 150 Hz and 300 V) except for 3 out of 449 turkeys that were found conscious at the exit of the waterbath due to lack of contact with the electrified water. Similarly occurred in turkeys approximately 8 kg of body weight where a combination that resulted in effective stunning was found in SH-7 (sine alternative current (AC), 402 ± 50 mA/bird, 100 Hz and 330 V) were all animals were effectively stunned and none of them recovered consciousness before death. In turkeys with heavier body weights (approximately 15 kg), electrical parameters that were effective were found in SH-8 (sine alternative current (AC), 275 ± 57 mA/bird, 171 Hz and 243 ± 2 V) although few turkeys showed outcomes of consciousness both before and during bleeding. In this specific case, the ones that showed outcomes of consciousness were females within a batch of males in all cases. Females are shorter than males and missed the waterbath. SH-2 also deserves special attention since they did not adapt the key parameters to the sex and weight of the animals. Thus, using sine alternative current (AC), 300 ± 57 mA/bird, 199 Hz and 140 ± 2 V, the prevalence of recovery of consciousness increased dramatically as the average weight of the batch increased. Electrical combinations that strongly failed at inducing or maintaining unconsciousness were found when using higher frequencies (e.g., 330 and 400 Hz) such as in SH-6 and SH-4. However, most of the conscious turkeys found in SH-4 were males within a batch of females.

Males are heavier and taller than females, therefore, males found in a batch of females are more prone to be ineffectively stunned because the combination of electrical parameters might be adequate for the females (lower size and body weight) but not for the males. On the other hand, females found in a batch of males are more prone to miss the WB due to shorter stature. For that reason, better practices are needed to solve this issue that compromises bird welfare. Crating males separately from females on farm and then adjust both the height of the WB and the electrical parameters in use according to sex is proposed to maximize the stunning effectiveness and the welfare of turkeys at the time of killing. Other proposals include separate the misplaced animals and stun them with another flock that have similar characteristics or stun them directly with a backup method.

Additionally, it raises the possibility that failure in the maintenance of the equipment or poor or lack of calibration might be also responsible that the electrical parameters recorded differed from what is actually delivered in the WB. It might be the case that SH-5 was at the low frequency given (i.e., 75 Hz) the unexpectedly high prevalence of conscious turkeys.

## 5. Conclusions

Before bleeding, breathing is the most relevant indicator of consciousness. Others such as spontaneous blinking and vocalizations might be assessed as well, even if less present. Any turkey exhibiting at least one of these ABIs must be re-stunned with backup stunning equipment prior to bleeding. During bleeding, breathing and wing flapping are the most relevant ABIs of consciousness. Others such as head shaking might be assessed as well despite the low prevalence and low repeatability. Similar to before bleeding, any turkey exhibiting at least one outcome of consciousness at this stage must be re-stunned with backup stunning equipment as soon as detected. Furthermore, if several turkeys regain consciousness within the same batch, electrical parameters should be also readjusted to ensure that all turkeys are induced to unconsciousness and do not regain consciousness before death. On the other hand, it is of utmost importance to rely on animal-based indicators of the state of consciousness and not on the key electrical parameters applied in the waterbath.

## Figures and Tables

**Figure 1 animals-13-00668-f001:**
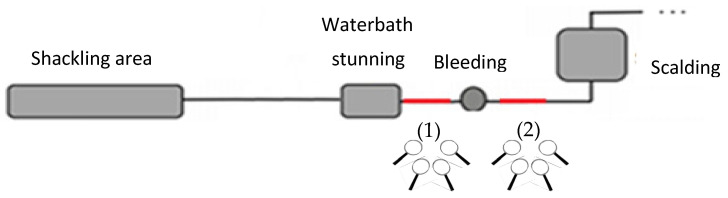
Position of the observers during the assessment of animal-based indicators of the effectiveness of waterbath stunning in turkeys. The position of the lens is the position of the observers (i.e., before (1) and during bleeding (2)) and the red segments are the observation area.

**Figure 2 animals-13-00668-f002:**
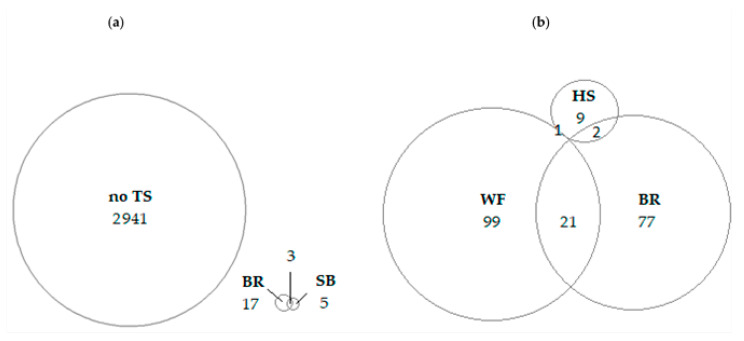
Venn diagram of the animal-based indicator of consciousness assessed in waterbath-stunned turkeys (**a**) before bleeding and (**b**) during bleeding. Indicators of consciousness are: no TS: absence of tonic seizure; BR: presence of breathing; SB: presence of spontaneous blinking; WF: presence of wing flapping; HS: presence of head shaking; SS: presence of spontaneous swallowing. Numbers specify the total amount of turkeys showing each indicator or combinations of indicators from a total of 3659 turkeys assessed before bleeding and 4218 during bleeding.

**Figure 3 animals-13-00668-f003:**
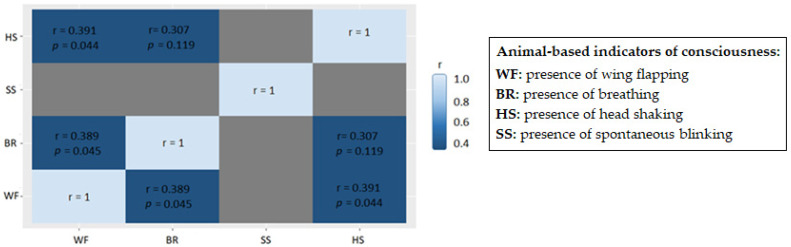
Heat map of correlations of the outcomes of the animal-based indicators of consciousness during bleeding in waterbath-stunned turkeys found when assessing 28 batches from eight different slaughterhouses. The values are Spearman’s rank correlation coefficients (r) and *p*-values. Grey areas mean that correlation could not be computed due to lack of presence of spontaneous swallowing.

**Table 1 animals-13-00668-t001:** Main characteristics of the eight slaughterhouses that used waterbath stunning systems included in this study.

	Slaughterhouse							
	1	2	3	4	5	6	7	8
WB length (m)	1.9	3.0	3.6	3.6	0.8	2.1	2.6	3.4
Birds in the WB (n)	5	4	8	5	2	7	7	10
Exposure time (s)	NA	9 (♂), 7 (♀)	12	18	6	22	13	22
Line speed (birds/h)	1000	2100 (♂), 3000 (♀)	1800	700	300	350	2000	1860
Stun-to-stick interval (s)	8–9	6	6	29	30	23	NA	13
Bleeding method	M	M	M	M	M	M	A	M

WB: waterbath, ♂: males, ♀: females; NA: not available; M: manually; A: automatically.

**Table 2 animals-13-00668-t002:** Number of batches of slaughtered turkeys, sex, average body weight and age of turkeys per batch for each slaughterhouse. The number of turkeys assessed before and during bleeding; the average electrical parameters ± standard deviation of the waterbath are reported.

		Characteristics of the Birds				No. Birds Assessed		Electrical Parameters		
SH	Batch	Sex	No. Birds	BW, kg	Age, d	BB	DB	Current, mA/Bird	Frequency, Hz	Voltage, V
1	1	Female	2000	4.2	57	0	200	289 ± 29	150	300
	2	Male	3148	3.8	64	199	200	345 ± 33	150	300
	3	Female	2770	3.8	64	250	300	367 ± 25	150	300
2	1	Male	1820	15.9	NA	0	39	293 ± 39	199	145 ± 0
	2	Male	4595	16.2	NA	187	150	319 ± 65	199	141 ± 2
	3	Female	2212	7.8	NA	0	190	303 ± 33	199	140 ± 0
	4	Female	2478	6.3	NA	192	68	278 ± 46	199	140 ± 0
3	1	Male	4500	16.6	126	199	200	NA	199	NA
	2	Male	5500	7.7	91	200	213	NA	199	NA
	3	Male	3456	15.1	127	181	180	NA	199	NA
	4	Male	4280	16.0	131	199	213	284 ± 50	196	187 ± 3
	5	Male	2050	11.1	113	69	180	287 ± 35	196	186 ± 8
	6	Male	1675	12.1	118	198	200	302 ± 55	196	206 ± 3
	7	Male	3880	15.6	124	246	200	311 ± 61	196	229 ± 2
4	1	Female	910	10.5	103	66	150	659 ± 66	400	375 ± 13
	2	Female	400	7.6	105	120	58	634 ± 49	400	360
	3	Female	400	10.8	105	0	142	677 ± 75	400	407 ± 5
5	1	Female	445	12.4	120	197	200	208 ± 88	75	285
	2	Female	462	12.4	122	99	148	177 ± 94	75	285
6	1	Female	126	7.6	80	50	90	479 ± 125	330	350
	2	Female	210	10.7	109	13	100	400 ± 119	330	350
	3	Female	400	10.5	111	137	128	241 ± 80	330	350
7	1	Female	2304	7.9	88	99	200	372 ± 26	100	330 ± 2
	2	Female	1992	8.5	99	217	87	408 ± 34	100	330 ± 3
	3	Female	2304	7.8	88	177	123	436 ± 47	100	330 ± 3
	4	Female	2304	7.7	88	164	0	388 ± 59	100	330 ± 3
	5	Female	1568	7.8	88	0	32	NA	NA	NA
8	1	Male	3500	15.4	128	200	224	275 ± 57	171	243 ± 2

SH: slaughterhouse; No. Birds: number of birds in the batch; BW: body weight; BB: before bleeding; DB: during bleeding; NA: not available.

**Table 3 animals-13-00668-t003:** Animal-based indicators (ABIs) assessed and descriptions of the outcomes of unconsciousness and consciousness in turkeys stunned by waterbath in two different stages: before and during bleeding.

Stage	ABI	Outcome of Unconsciousness	Outcome of Consciousness
Before bleeding	Tonic seizure	Bird shows general loss of muscle tone and a completely relaxed body and flaccid body, with no neck tension.	Bird shows arched and stiff neck (i.e., necks appear parallel to the ground) and wings held tightly close to the body.
	Breathing	Absence of movements of the beak or abdominal muscles around the cloaca associated to cessation of breathing.	Presence of either a minimum of two movements of the beak or abdominal muscles around the cloaca associated to breathing.
	Spontaneous blinking	Bird does not open/close eyelid on its own (fast or slow) without stimulation.	Bird opens/closes eyelid on its own (fast or slow) without stimulation.
	Vocalizations	Absence of single or repeated short and loud shrieking (screaming) at high frequencies.	Single or repeated shrieking (screaming).
During bleeding	Wing flapping	Absence of flapping with both wings.	Flapping with both wings and should not be confused with rapid trembling of the entire body of the bird.
	Breathing	Absence of movements of the beak or abdominal muscles around the cloaca associated to cessation of breathing.	Presence of either a minimum of two movements of the beak or abdominal muscles around the cloaca associated to breathing.
	Spontaneous swallowing	Absence of deglutition reflex.	Deglutition reflex triggered by water from the stunner or blood from the neck-cutting wound entering the mouth during bleeding.
	Head shaking	Bird does not shake its head from side to side.	Bird shakes its head from side to side to get rid of blood or water entering the nostrils.

**Table 4 animals-13-00668-t004:** Percentage of the outcomes of the animal-based indicators for the state of consciousness in turkeys after waterbath stunning but before bleeding according to the observer (A to D) and slaughterhouse assessed (1 to 8).

		Absence of TS, %						Presence of BR, %						Presence of SB, %						Presence of VC, %					
SH	Birds, n	A	B	C	D	Mean	*p*- Value	A	B	C	D	Mean	*p*- Value	A	B	C	D	Mean	*p*- Value	A	B	C	D	Mean	*p*- Value
1	449	100	100	100	100	100	1.000	0.4	0.7	0.7	0.7	0.6	0.965	0.2	0.0	0.0	0.0	0.0	0.381	0.0	0.0	0.0	0.0	0.0	-
2	379	15.8 ^a^	16.9 ^a^	26.9 ^b^	19.0 ^ab^	17.2	<0.001	0.0	0.3	0.5	0.0	0.2	0.299	1.6 ^b^	0.3 ^a^	0.0 ^a^	0.0 ^a^	0.1	0.003	0.0	0.0	0.0	0.0	0.0	-
3	1292	94.4 ^a^	98.3 ^b^	97.6 ^b^	98.1 ^b^	98.0	<0.001	1.0	1.4	2.1	1.4	1.5	0.142	0.1	0.2	0.0	0.1	0.1	0.572	0.0	0.0	0.0	0.0	0.0	-
4	186	100	100	100	100	100	1.000	0.0	0.0	0.5	0.0	0.1	0.391	0.0	0.5	0.0	0.0	0.1	0.391	0.0	0.0	0.0	0.0	0.0	-
5	296	91.2	91.6	94.6	92.9	92.6	0.385	7.8	7.4	11.1	8.1	8.6	0.347	0.3 ^ab^	1.7 ^b^	0.3 ^ab^	0.0 ^a^	0.2	0.037	0.3	0.0	0.7	0.0	0.3	0.299
6	200	98.5	99.5	98	98.5	98.5	0.626	0.0	0.0	1.0	0.0	0.2	0.111	0.0	0.0	0.0	0.0	0.0	-	0.0	0.0	0.0	0.0	0.0	-
7	657	53.4	51.6	57.1	54.9	54.2	0.232	0.6	0.5	0.5	0.3	0.5	0.880	0.2	0.2	0.0	0.2	0.2	0.801	0.0	0.0	0.0	0.0	0.0	-
8	200	98.5	98.5	98.0	97.5	98.1	0.862	1.5	1.0	1.0	1.0	1.0	0.953	0.0	1.0	0.5	0.5	0.5	0.570	0.0	0.0	0.0	0.0	0.0	-
All	3659	80.1 ^a^	81.3 ^ab^	83.2 ^b^	82.1 ^ab^	81.2	<0.001	1.2 ^a^	1.3 ^a^	2.0 ^b^	1.4 ^a^	1.4	0.029	0.3 ^ab^	0.3 ^b^	0.1 ^a^	0.1 ^a^	0.2	0.011	0.03	0.0	0.05	0.0	0.0	0.300

SH: slaughterhouse, n: number of birds, TS: tonic seizure; BR: breathing; SB: spontaneous blinking; VC: vocalization. ^a,b^ = values with different superscripts within the same raw differ among observers by chance when submitted to a chi-squared % defective test (*p* < 0.05).

**Table 5 animals-13-00668-t005:** Inter-observer proportion of agreement (PoA), Fleiss’ kappa coefficient (κ) and interpretation of the animal-based indicators for the state of consciousness before bleeding in turkeys according to the slaughterhouse assessed (1 to 8).

	Absence of TS			Presence of BR			Presence of SB			Presence of VC		
SH	PoA, %	κ (Interpretation)	*p*-Value	PoA, %	κ (Interpretation)	*p*-Value	PoA, %	κ (Interpretation)	*p*-Value	PoA, %	κ (Interpretation)	*p*-Value
1	100	*	-	99.8	0.00 (Poor)	0.511	99.8	0.00 (Poor)	0.511	100	*	-
2	58.6	0.28 (Poor)	<0.001	99.2	0.00 (Poor)	0.538	98.2	0.00 (Poor)	0.588	100	*	-
3	93.8	0.42 (Fair to good)	<0.001	98.4	0.72 (Fair to good)	<0.001	99.7	0.00 (Poor)	0.527	100	*	-
4	100	*	-	99.5	0.00 (Poor)	0.518	99.5	0.00 (Poor)	0.518	100	*	-
5	92.2	0.71 (Fair to good)	<0.001	93.6	0.77 (Excellent)	<0.001	97.6	0.00 (Poor)	0.599	99.0	0.00 (Poor)	0.543
6	97.5	0.48 (Fair to good)	<0.001	99.0	0.00 (Poor)	0.535	100	*	-	100	*	-
7	76.7	0.75 (Excellent)	<0.001	99.7	0.78 (Excellent)	<0.001	99.9	0.67 (Fair to good)	<0.001	100	*	-
8	96.0	0.43 (Fair to good)	<0.001	98.5	0.66 (Fair to good)	<0.001	99.0	0.50 (Fair to good)	<0.001	100	*	-
All	88.4	0.79 (Excellent)	<0.001	98.6	0.75 (Excellent)	<0.001	99.4	0.15 (Poor)	<0.001	99.9	0.00 (Poor)	0.512

SH: slaughterhouse; TS: tonic seizure; BR: breathing; SB: spontaneous blinking; VC: vocalization. * Insufficient scoring variation to calculate κ (all indicator scores were 0). κ interpretation: ≥0.75 “excellent”, 0.40–0.74 “fair to good”, and <0.40 “poor” agreement [20].

**Table 6 animals-13-00668-t006:** Percentage of the outcomes of the animal-based indicators for the state of consciousness in waterbath-stunned turkeys during bleeding according to observer (A to D) and slaughterhouse assessed (1 to 8).

		Presence of WF, %						Presence of BR, %						Presence of SS, %						Presence of HS, %					
SH	Birds, n	A	B	C	D	Mean	*p*-Value	A	B	C	D	Mean	*p*-Value	A	B	C	D	Mean	*p*-Value	A	B	C	D	Mean	*p*-Value
1	700	0.0	0.0	0.0	0.0	0.0	-	0.0	0.0	0.0	0.0	0.0	-	0.0	0.0	0.0	0.0	0.0	-	0.0	0.0	0.0	0.0	0.0	-
2	447	5.4	5.1	4.7	4.9	5.0	0.972	6.3	4.9	4.9	3.8	5.0	0.412	0.2	0.0	0.0	0.0	0.0	0.391	0.9	0.9	1.6	0.7	1.0	0.568
3	1389	3.4	2.5	3.1	2.4	2.9	0.380	0.8	0.9	0.9	0.6	0.8	0.682	0.0	0.0	0.0	0.0	0.0	-	0.1	0.4	0.1	0.1	0.1	0.188
4	350	3.7	3.7	3.4	3.1	2.6	0.972	6.9	6.9	6.9	6.3	6.7	0.987	0.0	0.0	0.0	0.0	0.0	-	0.3	0.9	0.3	0.0	0.4	0.282
5	348	3.7	4.0	3.7	3.4	2.8	0.984	1.7	1.7	2.3	1.7	1.9	0.925	0.0	0.0	0.0	0.0	0.0	-	0.0	0.0	0.3	0.0	0.1	0.391
6	318	2.2	1.3	2.2	2.2	1.5	0.777	9.1	8.5	8.8	8.8	8.8	0.994	0.0	0.0	0.0	0.0	0.0	-	0.0	0.3	0.0	0.0	0.1	0.391
7	442	0.2	0.2	0.2	0.2	0.4	-	1.1	1.1	1.1	0.9	1.1	0.984	0.0	0.0	0.0	0.0	0.0	-	0.2	0.2	0.0	0.0	0.1	0.572
8	224	1.3	0.4	0.4	0.4	0.9	0.570	0.9	0.4	0.4	0.4	0.4	0.896	0.0	0.0	0.0	0.0	0.0	-	0.0	0.0	0.0	0.0	0.0	-
All	4218	2.6	2.2	2.3	2.1	2.3	0.472	2.5	2.3	2.4	2.0	2.3	0.550	0.0	0.0	0.0	0.0	0.0	-	0.2	0.3	0.2	0.1	0.2	0.248

SH: slaughterhouse, n: number of birds, WF: wing flapping; BR: breathing; SS: spontaneous swallowing; HS: head shaking. No value within the same raw differ among observers by chance when submitted to a chi-squared % defective test (*p* > 0.05).

**Table 7 animals-13-00668-t007:** Inter-observer proportion of agreement (PoA), Fleiss’ kappa coefficient (κ) and its interpretation of the animal-based indicators for the state of consciousness during bleeding according to the slaughterhouse assessed (1 to 8).

	Presence of WF			Presence of BR			Presence of SS			Presence of HS		
SH	PoA, %	κ (Interpretation)	*p*-Value	PoA, %	κ (Interpretation)	*p*-Value	PoA, %	κ (Interpretation)	*p*-Value	PoA, %	κ (Interpretation)	*p*-Value
1	100	*	-	100	*	-	100	*	-	100	*	-
2	94.9	0.70 (Fair to good)	<0.001	95.1	0.73 (Fair to good)	<0.001	99.8	0.00 (Poor)	0.512	97.8	0.40 (Fair to good)	<0.001
3	98.1	0.81 (Excellent)	<0.001	98.6	0.54 (Fair to good)	<0.001	100	*	-	99.6	0.30 (Poor)	<0.001
4	98.3	0.85 (Excellent)	<0.001	96.9	0.87 (Excellent)	<0.001	100	*	-	98.6	0.00 (Poor)	0.513
5	97.7	0.84 (Excellent)	<0.001	98.9	0.82 (Excellent)	<0.001	100	*	-	99.7	0.00 (Poor)	0.513
6	99.0	0.88 (Excellent)	<0.001	98.7	0.95 (Excellent)	<0.001	100	*	-	99.7	0.00 (Poor)	0.514
7	100	*	-	99.8	0.95 (Excellent)	<0.001	100	*	-	99.8	0.33 (Poor)	<0.001
8	99.1	0.75 (Excellent)	<0.001	99.3	0.57 (Fair to good)	<0.001	100	*	-	100	*	-
All	98.4	0.80 (Excellent)	<0.001	98.5	0.82 (Excellent)	<0.001	99.9	0.00 (Poor)	0.504	99.5	0.29 (Poor)	<0.001

SH: slaughterhouse; WF: wing flapping; BR: breathing; SS: spontaneous swallowing; HS: head shaking. * Insufficient scoring variation to calculate κ (all indicator scores were 0). κ interpretation: ≥0.75 “excellent”, 0.40–0.74 “fair to good”, and <0.40 “poor” agreement [20].

**Table 8 animals-13-00668-t008:** Number of turkeys assessed (n), average body weight (BW), electrical parameters used in the waterbath, and the mean prevalence and 95% confidence interval (CI) of turkeys showing at least one outcome of consciousness before bleeding according to the slaughterhouse (SH; 1 to 8) and batch assessed.

					Electrical Parameters in Waterbath			Turkeys with Outcomes of Consciousness	
SH	Batch	Birds, n	Sex	BW, kg	Current, mA/Bird	Frequency, Hz	Voltage, V	Mean, % ¥	95% CI
1	2	199	Male	3.8	345 ± 33	150	300	1.5	[0.5–4.3]
1	3	250	Female	3.8	367 ± 25	150	300	0.0	[0.0–1.5]
2	2	187	Male	16.2	319 ± 65	199	141 ± 2	0.7	[0.1–3.0]
2	4	192	Female	6.3	278 ± 46	199	140 ± 0	0.0	[0.0–1.9]
3	1	199	Male	16.6	NA	199	NA	0.8	[0.1–2.8]
3	2	200	Male	7.7	NA	199	NA	0.6	[0.1–2.8]
3	3	181	Male	15.1	NA	199	NA	1.2	[0.3–3.9]
3	4	199	Male	16.0	284 ± 50	196	187 ± 3	1.5	[0.5–4.3]
3	5	69	Male	11.1	287 ± 35	196	186 ± 8	1.4	[0.3–7.8]
3	6	198	Male	12.1	302 ± 55	196	206 ± 3	2.0	[0.8–5.1]
3	7	246	Male	15.6	311 ± 61	196	229 ± 2	2.8	[1.4–5.8]
4	1	66	Female *	10.5	659 ± 66	400	375 ± 13	0.0	[0.0–5.5]
4	2	120	Female *	7.6	634 ± 49	400	360	0.4	[0.1–4.6]
5	1	197	Female *	12.4	214 ± 88	75	285	5.8	[3.5–10.3]
5	2	99	Female *	12.4	177 ± 94	75	285	15.9	[10.2–24.7]
6	1	50	Female	7.6	479 ± 125	330	350	0.5	[0.0–7.1]
6	2	13	Female	10.7	400 ± 119	330	350	0.0	[0.0–22.8]
6	3	137	Female	10.5	241 ± 80	330	350	0.0	[0.0–2.7]
7	1	99	Female	7.9	372 ± 26	100	330 ± 2	0.0	[0.0–3.7]
7	2	217	Female	8.5	408 ± 34	100	330 ± 3	0.0	[0.0–1.7]
7	3	177	Female	7.8	436 ± 47	100	330 ± 3	0.0	[0.0–2.1]
8	1	200	Male **	15.4	275 ± 57	171	243 ± 2	1.0	[0.3–3.6]

¥: mean of the four observers; NA: not available; * mostly females but some males; ** mostly males but some females.

**Table 9 animals-13-00668-t009:** Number of turkeys assessed (n), average body weight (BW), electrical parameters used in the waterbath, and the mean prevalence and 95% confidence interval (CI) of turkeys showing at least one outcome of consciousness during bleeding according to the slaughterhouse (SH; 1 to 8) and batch assessed.

					Electrical Parameters in Waterbath			Turkeys with Outcomes of Consciousness	
SH	Batch	Birds, n	Sex	BW, kg	Current, mA/Bird	Frequency, Hz	Voltage, V	Mean, % ¥	95% CI
1	1	200	Female	4.2	289 ± 29	150	300	0.0	[0.0–1.9]
1	2	200	Male	3.8	345 ± 33	150	300	0.0	[0.0–1.9]
1	3	200	Female	3.8	367 ± 25	150	300	0.0	[0.0–1.3]
2	1	39	Male	15.9	293 ± 39	199	145 ± 0	10.8	[4.1–23.6]
2	2	150	Male	16.2	319 ± 65	199	141 ± 2	17.6	[12.7–24.9]
2	3	190	Female	7.8	303 ± 33	199	140 ± 0	6.6	[4.1–11.4]
2	4	68	Female	6.3	278 ± 46	199	140 ± 0	2.6	[0.8–10.1]
3	1	200	Male	16.6	NA	199	NA	1.4	[0.5–4.3]
3	2	213	Male	7.7	NA	199	NA	4.3	[2.2–7.8]
3	3	180	Male	15.1	NA	199	NA	3.6	[1.9–7.8]
3	4	213	Male	16.0	284 ± 50	196	187 ± 3	2.5	[1.0–5.4]
3	5	180	Male	11.1	287 ± 35	196	186 ± 8	2.0	[1.0–5.6]
3	6	200	Male	12.1	302 ± 55	196	206 ± 3	8.5	[3.1–9.6]
3	7	200	Male	15.6	311 ± 61	196	229 ± 2	8.5	[3.1–9.6]
4	1	150	Female *	10.5	659 ± 66	400	375 ± 13	7.0	[4.1–12.6]
4	2	58	Female *	7.6	634 ± 49	400	360	1.7	[0.3–9.1]
4	3	142	Female *	10.8	677 ± 75	400	407 ± 5	9.0	[5.4–15.0]
5	1	200	Female *	12.4	214 ± 88	75	285	5.8	[3.5–10.2]
5	2	148	Female *	12.4	177 ± 94	75	285	5.1	[2.8–10.3]
6	1	90	Female	7.6	479 ± 125	330	350	8.6	[4.6–16.6]
6	2	100	Female	10.7	400 ± 119	330	350	8.0	[4.1–15.0]
6	3	121	Female	10.5	241 ± 80	330	350	10.4	[6.4–17.5]
7	1	200	Female	7.9	372 ± 26	100	330 ± 2	1.0	[0.3–3.6]
7	2	87	Female	8.5	408 ± 34	100	330 ± 3	0.0	[0.0–4.3]
7	3	123	Female	7.8	436 ± 47	100	330 ± 3	0.0	[0.0–3.0]
8	1	224	Male **	15.4	275 ± 57	171	243 ± 2	1.3	[0.5–3.9]

¥: mean of the four observers; NA: not available; * mostly females but some males; ** mostly males but some females.

## Data Availability

Data are contained within this article.

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
