# Peer review of "Relevant Indicators of Consciousness after Waterbath Stunning in Turkeys and Stunning Efficiency in Commercial Conditions"

_animals, 2023, doi:10.3390/ani13040668_

Round 1

Reviewer 1 Report

General comments:

 The manuscript animals-2118242, entitled "Inter-observer Repeatability of Indicators of Consciousness after Waterbath Stunning in Turkeys" with Ms. Contreras-Jodar as first author deals with the important topic of incorrect stunning in turkey slaughterhouses. The manuscript is relatively structured, carefully formulated and presents interesting ideas. However, some sections need to be revised and major mistakes must be corrected especially in the results and discussion section. Various comments for changes to the text could be find in the following section.

Detailed comments:

 L2: The prevalences you have determined are very interesting and should therefore also be mentioned in the title.

 L5: Please delete the full stop after “Spain”

 L10: Simple summary and Abstract are in a smaller font size. Is this required by the journal?

L15: Maybe it is better to use the term “standardization” in this sentence?

L24: The first sentence should be placed at the end of the abstract, because it seems to be the conclusion of this study.

L30: How can correlations provide insights regarding reliability? Reliability depends primarily on the assessors.

L34: How did you choose this list? Please provide a reason for the indicators you sorted out. In which respect did they have disadvantages?

L37: Do the keywords have to be listed in alphabetical order?

L64: Can you say something more about the CAS method? Which gases are normally used? What possibly negative effects are to be expected on the animals? Respiratory distress, suffocation? Why are the animals not reversibly stunned by this method?

L70: Can you provide information on the prevalences of non-stunned animals using the WBS method?

L80: This sentence is too long. Please split into two sentences (i.e. poultry species / repeatability); Please correct: “species-specific”

L92: Please correct: “species”

L98: How can correlations provide insights regarding reliability? Reliability depends primarily on the assessors.

L99: I would add the word "we" here (after “offer”)

L99: I would add “and reliable” (after “relevant”)

Table 1: Please use the same font in the table as in the main text

Table 1: Why were sometimes the females and sometimes the males left in the water bath longer (SH 2 vs. SH3)? Do you have an explanation why the slaughterhouses handle this differently? Or was that a typing error?

Table 1: The stun-to-stick interval is in some slaughterhouses relatively long (30 sec). What is the maximum time allowed? When do the animals normally start to wake up again?

L120: For SH 4, no gender-specific values were given at all?

L125: Maybe better “...they are shown…”?

L131: Please move the heading to the left margin and use italic font.

L133: Please provide more information on the background of the assessors. What training and experience do they have in assessing animals? How and with what success have they been trained? Were assessor agreement tests conducted prior to the surveys? What were the results, if any?

Figure 1: The graphics are a bit out of focus. Could you paste it in a better quality?

Figure 1: Maybe it makes sense to insert (1) and (2) in the graphic?

Figure 1: Were there size differences between the assessors? How close were the assessors and could the respective location have an influence on the visibility of the indicators?

Figure 1: Figures and tables should be self-explanatory, so please do not use abbreviations here.

L163: Please correct the formatting of the heading.

L169: Please insert the name of the author [16]

Table 2: Please try to display the table 2 on one page

Table 2: Please use the same font in the table as in the main text

Table 2: Why were the turkeys at SH 1 slaughtered so young and with low weight?

Table 2: Why were the electrical parameters not adjusted in some slaughterhouses? Surely it matters whether heavier males or lighter females are slaughtered, right?

Table 3: Please highlight the modifications you have made compared to the original EFSA indicators (e.g. bold letters)

Table 3: How is it possible to pay attention to the beak and the muscles around the cloaca at the same time (indicator "breathing")? Was the speed reduced for the examinations?

L200: Another way to calculate the assessor agreement is the prevalence-adjusted, bias-adjusted kappa (PABAK). In this formula, the existing prevalences are taken into account. Why did you not use this one? Please justify your choice in more detail.

L277: There are missing words in this sentence: Therefore, the calculated…

Table 4: Please insert (1 to 8) after “slaughterhouses” in the heading

Table 4: Tables should be self-explanatory, so please mention again in the heading the statistical test used

Table 4: “p-value” instead of “p-Value”

Table 4: Please delete the first “0” in the value “09.0”

Table 4: Please harmonize within the manuscript: Only one decimal places after the comma (except Kappa and p-values)!

Table 4: Please write the footnotes closer to the table

Table 4: Why are the total scores of assessors A and D regarding TS significantly different, but A and C are not?

Table 4: How is it possible that the overall values regarding BR are significant but the individual values are not?

Table 5: Please write the footnotes closer to the table

Table 5: Table 5 would be easier to interpret if rows and columns were analogous to Table 4. Could you please standardize the layout between the tables?

Table 6: Maybe it makes sense to combine tables 4 and 6 to save space?

Table 6: Tables should be self-explanatory, so please mention again in the heading the statistical test used

Table 6: Please insert (1 to 8) after “slaughterhouses” in the heading

Table 6: There is missing a comma after “HS”

Table 7: Table 7 would be easier to interpret if rows and columns were analogous to Table 6. Could you please standardize the layout between the tables?

Table 7: “Presence” instead of “Presnece”

Table 7: Please write the footnotes closer to the table

All tables: The page numbers are not correct for the tables.

Figure 2: The graphics are a bit out of focus. Could you paste it in a better quality?

Figure 2: Presence of vocalization is not shown in the figure. Therefore it should not be included in the heading. Alternatively you could present the indicator as a point (i.e. zero animals).

Figure 2: It is confusing that the circle for "WF" is almost the same size as the one for "no TS". Could you reproduce the circles in the correct proportions?

Figure 3: It is not necessary to highlight the non-significant results with a red cross. The significance can be checked by looking at the p-values. Therefore, please remove the crosses.

Figure 3: For the assessment of the correlation, it is not so much the significance, but rather the level of the correlation coefficient that is decisive. Please explicitly point out that the correlations are rather weak.

Figure 3: It should be mentioned what the gray fields mean.

L404: In my opinion, the discussion is far too brief and contains few comparisons with other studies. Please expand this chapter to include other publications on the subject.

L416: Please also go into more detail about each person and their experience in evaluating animals. Were there any physical differences that contributed to the divergent results?

L435: Can you provide references from other studies as a comparison? How high were the assessor agreements there?

L465: Perhaps it is also necessary to reduce the slaughterline speed? Or to improve the lighting conditions?

L484: It would be very interesting to investigate whether the different parameters in the slaughterhouses (e.g. current intensity) affect the number of unstunned animals. Could you do some analysis for this? It would certainly increase the quality of the manuscript.

L484: I would add more detail to the discussion on the prevalences of unstunned animals that have been recorded. Are these within the permitted range or should the slaughterhouses optimize the processes? Could you provide comparative values from other studies or in other species?

L485: From my point of view, it would make more sense to integrate the recommendations into the discussion and provide further references. The conclusions should then contain only the most important findings.

L528: The font sizes differ in the reference list. Please standardize the layout according to the specifications of the journal.

L528: The journal names differ in the reference list (with or without full stops). Please standardize the layout according to the specifications of the journal.

L567: Why (Basel)?

Reviewer 2 Report

This is a very well presented study on an important topic. The work has been completed with scientific rigor and appropriate methods. I have only minor comments.

Lines 33-36 In the last sentences of the abstract, I find the use of the words ‘neglected’ rather strange. It would be clearer just to state what the recommended indicators are.  It would be useful to state if this was based on their reliability, validity, or both.

Lines 44-46 It would be useful to have sense of the proportions of turkeys stunned with each method, if this information is available.

Lines 81 and 92 Use species instead of specie.

Round 2

Reviewer 1 Report

General comments:

The manuscript animals-2118242, entitled "Relevant Indicators of Consciousness after Waterbath Stunning in Turkeys and Stunning Efficiency in Commercial Conditions" with Ms. Contreras-Jodar as first author was properly revised by the authors. The reviewer comments were considered in the revision and the authors left none of my questions unanswered. However, some minor inconsistencies and logical mistakes can still be found in the manuscript. The suggested changes (see detailed comments) should be implemented by the authors in a second revision. Various comments for suggested changes to the text could be find in the following section (Detailed comments).

Detailed comments:

General note: Changes in the revision should be made using the track changes function of Word to make them easier to follow.

L26: The abstract is sometimes read alone (e.g. in literature databases), so abbreviations should be written out again here (WBS).

L39: Please still include the average prevalence values and possibly min and max in the abstract.

Table 1: The heading of Table 1 is formatted differently compared to the other tables. Please adjust the format.

Table 1: What does the asterisk stand for in “bleeding method”?

L131: Please add here the reason for the higher line speed in the females.

Figure 1: Can you use the same font for the caption of the figure as in the text?

L146: Please add here that the definition of the parameters do not differ significantly between broilers and turkeys.

L200: Another way to calculate the assessor agreement is the prevalence-adjusted, bias-adjusted kappa (PABAK). In this formula, the existing prevalences are taken into account. Why did you not use this one? Please justify your choice in more detail.

Main restriction for PABAK is that it only can be computed to examine the agreement between two observers as it is referenced in O’Leary et al. (2014 https://doi.org/10.1016/j.physio.2013.08.002). In our study, we were four observers and therefore, PABAK was not possible… In any case, many thanks for your suggestion because we did not know about PABAK and who knows if we will use it in the future. It is always nice to learn.

The PABAK can actually be used with multiple assessors. For this purpose, the assessors are compared individually with a so called "silver standard". The silver standard is usually the person with the most experience in using the assessment system.

L225: Please harmonize within the manuscript: “-1” or “−1”!

L309: In this sentence, some words must have been deleted by mistake (Therefore, …).

L324: Please harmonize within the manuscript: ‘excellent’ or “excellent”.

Table 6: First column; “All” instead of “all”.

Table 6: Why is it “Absence of WF” and not “Presence of WF”?

Figure 2: Can you use the same font for the caption of the figure as in the text?

Figure 2: Please harmonize within the manuscript: “2941” or “2,941”.

L417: Were there 3459 or 3659 turkeys that were assessed? Different information can be found in the manuscript (e.g. Table 4).

Figure 3: Can you use the same font for the caption of the figure as in the text?

L469: No data from batch 1 in Slaughterhouse 1 are shown in the table. Do you perhaps mean batch 3?

L470: It is not effective stunning, if any turkey showed indicators of consciousness. Please correct this sentence.

L494: The discussion has become much more comprehensive and understandable. I would still like you to include more comparative studies. Even if there are only a few publications with the same topic, you could also include papers with related topics as a reference (e.g. waterbath stunning in poultry).

L672: Please provide reasons for this statement (e.g. prevalences of unstunned animals are sometimes unacceptably high).

L695: Please harmonize within the manuscript: “And” between two authors or not?

L697: There is missing a blanket space between “stunning” and “of”.

L699: Please harmonize within the manuscript: “Poult Sci 2010a” or “Poult Sci. 2010a”!

L707: The author names are not formatted correctly.
